# Ab initio calculation of real solids via neural network ansatz

Xiang Li [1] ✉, Zhe Li [1] & Ji Chen [2]

Neural networks have been applied to tackle many-body electron correlations for small molecules and physical models in recent years. Here we propose an architecture that extends molecular neural networks with the inclusion of periodic boundary conditions to enable ab initio calculation of real solids. The accuracy of our approach is demonstrated in four different types of systems, namely the one-dimensional periodic hydrogen chain, the two-dimensional graphene, the three-dimensional lithium hydride crystal, and the homogeneous electron gas, where the obtained results, e.g. total energies, dissociation curves, and cohesive energies, reach a competitive level with many traditional ab initio methods. Moreover, electron densities of typical systems are also calculated to provide physical intuition of various solids. Our method of extending a molecular neural network to periodic systems can be easily integrated into other neural network structures, highlighting a promising future of ab initio solution of more complex solid systems using neural network ansatz, and more generally endorsing the application of machine learning in materials simulation and condensed matter physics.

Solving the many-body electronic structure of real solids from ab initio is one of the grand challenges in condensed matter physics and materials science[1]. More accurate ab initio solutions can push the limit of our understanding of many fundamental and mysterious emergent phenomena, such as superconductivity, light–matter interaction, and heterogeneous catalysis, to name just a few[2]. The current workhorse method is density functional theory (DFT), whose accuracy depends quite sensitively on the choice of the so-called exchange-correlation functional and unfortunately there lacks a systematic routine towards the exact[3,4]. Other commonly used ab initio quantum chemistry methods, such as the coupled-cluster and configuration interaction theories[5], can provide more accurate solutions for molecules but face severe difficulty when applied to solid systems due to their high computational complexity. Recently, several breakthroughs have been made in applying these quantum chemistry methods on solids[6,7], driving the study of solid systems towards a new frontier.

Meanwhile, in the last few years, many attempts to tackle the correlated wavefunction problem in molecules or model Hamiltonians

using neural network-based approaches have been reported by different groups[8–16]. The key idea is to use the neural network as the wavefunction ansatz in quantum Monte Carlo (QMC) simulations. The stochastic nature of QMC enables a considerably economical time scaling and efficient parallelization[6,17–19]. The universal approximation theorem behind neural network-based ansatz significantly improves the accuracy of the traditional QMC method. This strategy has been proved successful in studying small molecules[10–13] in the first and second quantization, and solids in the second quantization[14]. However, how to apply such neural network ansatz for real solids in continuous space, and whether it can describe the long-range electron correlations in extended systems remain as open questions.

Here we propose a powerful periodic neural network ansatz for solids, which combines periodic distance features[20] with existing molecule neural networks[10]. Based on that, we develop a highly efficient QMC method for ab initio calculation of real solid and general periodic systems with high accuracy. We apply our method to periodic hydrogen chains, graphene, lithium hydride (LiH) crystals, and

[1]ByteDance Inc, Zhonghang Plaza, No. 43, North 3rd Ring West Road, Haidian District, Beijing, China. [2]School of Physics, Interdisciplinary Institute of Light-Element Quantum Materials, Frontiers Science Center for Nano-Optoelectronics, Peking University, Beijing 100871, P. R. China.
✉ e-mail: lixiang.62770689@bytedance.com

homogeneous electron gas. These systems cover a wide range of interests, including materials dimension from one to three, electronic structures from metallic to insulating, and bonding types from covalent to ionic. Standard techniques are employed to reduce finite-size errors. The calculated dissociation curve, cohesive energy and correlation energy, can be compared satisfactorily with available experimental values and other state-of-the-art computational approaches. Electron densities of typical systems are further calculated to test our neural network and explore the underlying physics. All the results demonstrate that our method can achieve accurate electronic structure calculations of solid/periodic systems. In parallel to our work, refs. 21, 22 also developed periodic versions of neural networks to study the homogeneous electron gas system and obtained high-accuracy results. A more detailed comparison is discussed in the following sections.

## Results

### Neural network for a solid system

Periodicity and anti-symmetry are two fundamental properties of the wavefunction of a solid system. The anti-symmetry can be ensured by the Slater determinant, which has been commonly used as the basic block in molecular neural networks. We also approximate the wavefunction by two Slater determinants of one spin-up channel and one spin-down channel,

$$\psi(\mathbf{r}) = \mathrm{Det}_\uparrow\left[e^{i\mathbf{k}\cdot\mathbf{r}_\uparrow}u_{\mathrm{mol}}^\uparrow(d)\right]\mathrm{Det}_\downarrow\left[e^{i\mathbf{k}\cdot\mathbf{r}_\downarrow}u_{\mathrm{mol}}^\downarrow(d)\right]. \quad (1)$$

In this regard, our ansatz resembles the structure of FermiNet[10,11], whereas other neural network wavefunction ansatz may include extra terms in addition to the Slater determinants[12]. Each determinant is then constructed from a set of periodic orbitals, which inherits the physics captured by the generalized collective Bloch function formed by a product of phase factor $e^{i\mathbf{k}\cdot\mathbf{r}}$ and collective molecular orbital $u_{\mathrm{mol}}$. The generalized many-body Bloch function incorporates electron correlations and goes beyond single-electron approximation[18].

Figure 1 displays more details on the structure of our neural network. Building an efficient and accurate periodic ansatz is the key step in developing ab initio methods for solids. Here we have followed the recently proposed scheme of Whitehead et al. to construct a set of periodic distance features $d(\mathbf{r})$[20] using lattice vectors in real and reciprocal space $(\mathbf{a}_i, \mathbf{b}_i)$,

$$d(\mathbf{r}) = \frac{\sqrt{\mathbf{A}\mathbf{M}\mathbf{A}^T}}{2\pi}, \mathbf{A} = (\mathbf{a}_1, \mathbf{a}_2, \mathbf{a}_3),$$
$$\mathbf{M}_{ij} = f^2(\omega_i)\delta_{ij} + g(\omega_i)g(\omega_j)(1 - \delta_{ij}), \omega_i = \mathbf{r}\cdot\mathbf{b}_i. \quad (2)$$

The periodic metric matrix $\mathbf{M}$ is constructed by periodic functions $f, g$, which retain ordinary distances at the origin and regulate them to periodic ones at far distances, ensuring asymptotic cusp form, continuity, and periodicity requirement at the same time.

The constructed periodic distance features $d(\mathbf{r})$ can then be fed into molecular neural networks to form collective orbitals $u_{\mathrm{mol}}$. Specifically, in this work, we represent the molecular networks with FermiNet[10], which incorporates electron-electron interactions. The inclusion of all-electron features promotes the traditional single-particle orbitals to the collective ones, and hence the description of wavefunction and correlation effects can be improved while fewer Slater determinants are required. In addition, the wavefunction of solid systems is necessarily complex-valued, and we introduce two sets of molecular orbitals to represent the real and imaginary parts of the solid wavefunction, respectively. The plane-wave phase factors $e^{i\mathbf{k}\cdot\mathbf{r}}$ in Fig. 1 are used to construct the Bloch function-like orbitals, and the corresponding $\mathbf{k}$ points are selected to minimize the Hartree-Fock (HF) energy.

Based on the variational principle, our neural network is trained using the variational Monte Carlo (VMC) approach. To efficiently optimize the network, a Kronecker-factored curvature estimator (KFAC) optimizer[23] implemented by DeepMind team[24] is modified and adopted, which significantly outperforms traditional energy minimization algorithms. Calculations are also ensured by efficient and massive parallelization on multiple nodes of high-performance GPUs. More details on the theories, methods, and computations are included in the Methods section and the supplementary information.

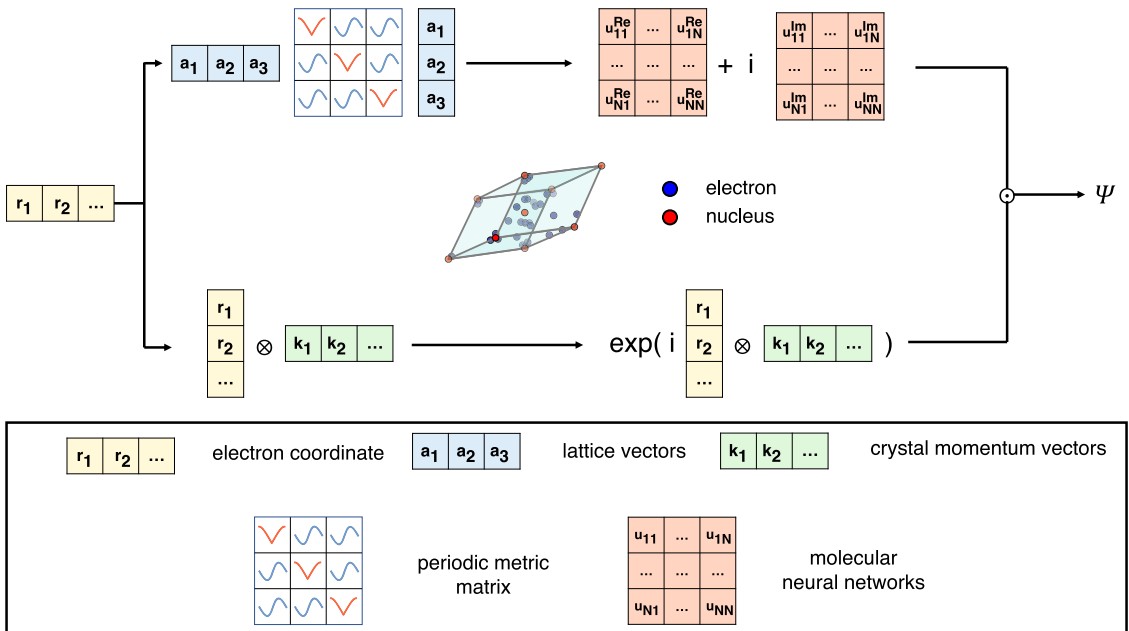

**Fig. 1 | Sketch of neural network architecture.** The electron coordinates $\mathbf{r}_i$ are passed to two channels. In the first one, they build the periodic distance features $d(\mathbf{r})$ using the periodic metric matrix $\mathbf{M}$ and the lattice vectors $\mathbf{a}$, and then $d(\mathbf{r})$ features are fed into two molecular neural networks, that represent separately the real and the imaginary part of the wavefunction. In the second channel, $\mathbf{r}_i$ constructs the plane-wave phase factors on a selected set of crystal momentum vectors. The total wavefunctions of solids are constructed by the two channels following the expression of Eq. (1).

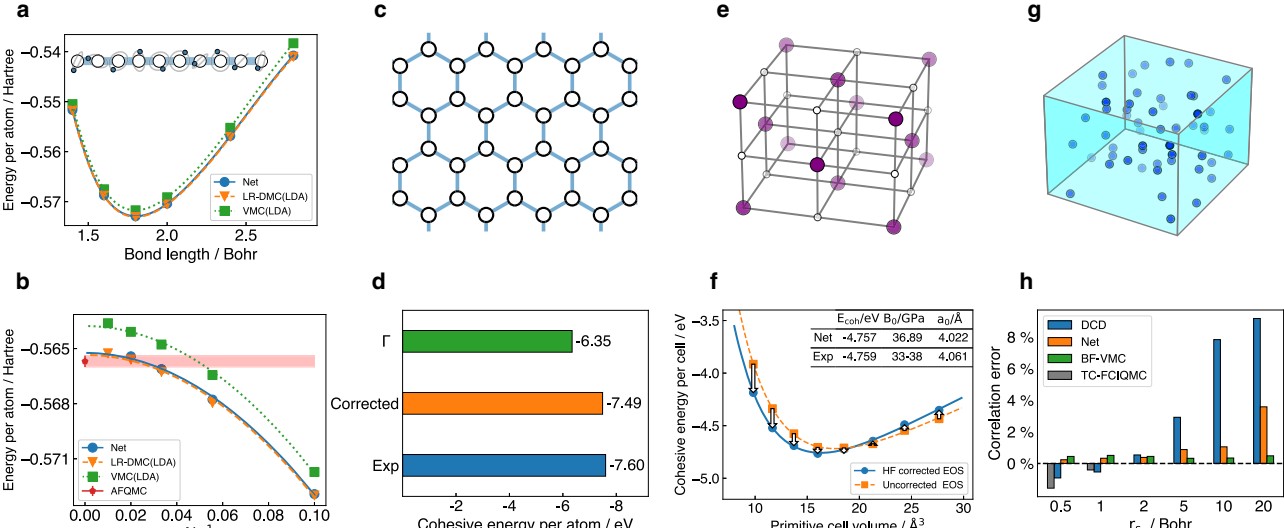

**Fig. 2 | Calculated results of neural network.** Our results are all labeled as Net. Statistical errors are negligible for the presented data. **a** $H_{10}$ dissociation curve is plotted. **b** energy of different hydrogen chain sizes N, the bond length of the hydrogen chain is fixed at 1.8 Bohr. LR-DMC and VMC both use the cc-pVTZ basis set, and the one-body Jastrow function uses orbitals from LDA calculations. AFQMC is pushed to complete the basis limit. All the comparison results are taken from ref. 25. **c** Structure of graphene. **d** the cohesive energy per atom of Γ point and finite-size error corrected result is plotted. Experiment cohesive energy is from ref. 29. Graphene is calculated at its equilibrium length 1.421 Å. **e** Structure of rock-salt lithium hydride crystal. **f** Equation of state of LiH crystal is plotted, fitted Birch–Murnaghan parameters and experimental data are also given. HF corrections are calculated using cc-pVDZ basis, and $E_\infty^{HF}$ is approximated by $E_{N=8}^{HF}$. The arrows denote the corresponding HF corrections. **g** Plot of homogeneous electron gas system. **h** Correlation error of 54-electrons HEG systems at different $r_s$. Correlation error is defined as $[1 - (E - E_{HF})/(E_{ref} - E_{HF})] \times 100\%$, and $E_{HF}$ is taken from ref. 33. DCD, BF-VMC, and TC-FCIQMC results are displayed for comparison, and BF-DMC data were used as reference[33,34].

## Hydrogen chain

Hydrogen chain is one of the simplest models in condensed matter research. Despite its simplicity, the hydrogen chain is a challenging and interesting system, serving as a benchmark system for electronic structure methods and featuring intriguing correlated phenomena[25]. The calculated energy of the periodic $H_{10}$ chain as a function of the bond length is shown in Fig. 2a. The results from lattice-regularized diffusion Monte Carlo (LR-DMC) and traditional VMC are also plotted for comparison[25]. We can see that our results nearly coincide with the LR-DMC results and significantly outperform traditional VMC (see Supplementary Table 3). In Fig. 2b, the energy of hydrogen chains of different atom numbers are calculated for extrapolation to the thermodynamic limit (TDL). The shaded bar in Fig. 2b illustrates the extrapolated energy of the periodic hydrogen chain at TDL from auxiliary field quantum Monte Carlo (AFQMC), which is considered as the current state-of-the-art along with LR-DMC. Our TDL result is comparable with both AFQMC and LR-DMC (see Supplementary Table 4).

## Graphene

Graphene is arguably the most famous two-dimensional system (Fig. 2c) receiving broad attention in the past two decades for its mechanical, electronic, and chemical applications[26]. Here we carry out simulations to estimate its cohesive energy, which measures the strength of C-C chemical bonding and long-range dispersion interactions. The calculations are performed on a 2 × 2 supercell of graphene using twist average boundary condition (TABC)[27] in conjunction with structure factor $S(\mathbf{k})$ correction[28] (see Supplementary Fig. 3) to reduce the finite-size error. The calculated results are plotted in Fig. 2d along with the experimental value[29], and it shows that our neural network can deal with graphene very well, producing a cohesive energy of graphene within 0.1 eV/atom to the experimental reference (see Supplementary Table 6). We also plotted the results with periodic boundary conditions (PBC), namely the Γ point-only result, which deviates from the experiment data by 1.25 eV/atom.

## Lithium hydride crystal

For a three-dimensional system, we consider the LiH crystal with a rock-salt structure (Fig. 2e), another benchmark system for accurate ab initio methods[6,30,31]. Despite consisting of only simple elements, LiH represents typical ionic and covalent bonds upon changing the lattice constants. Using our neural network, we first simulate the equation of the state of LiH on a 2 × 2 × 2 supercell, as shown in Fig. 2f. In addition, we employ a standard finite-size correction based on Hartree–Fock calculations of a large supercell (see Supplementary Fig. 5). In Fig. 2f we also show the Birch–Murnaghan fitting to the equation of state, based on which we can obtain thermodynamic quantities such as the cohesive energy, the bulk modulus, and the equilibrium lattice constant of LiH. As shown in the inset, our results on the thermodynamic quantities agree very well with experimental data[30] (see Supplementary Table 8, 9).

For further validation, we have also computed directly the 3 × 3 × 3 supercell of LiH at its equilibrium length of 4.061 Å, which contains 108 electrons. To the best of our knowledge, this is the largest electronic system computed using a high-quality neural network ansatz. The 3 × 3 × 3 supercell calculation predicts the total energy per unit cell of LiH is −8.160 Hartree and the cohesive energy per unit cell is −4.770 eV after the finite-size correction (see Supplementary Table 10), which is also very close to the experimental value −4.759 eV[30].

## Homogeneous electron gas

In addition to the solids containing nuclei, our computational framework can also apply straightforwardly to model systems such as homogeneous electron gas (HEG). HEG has been studied for a long time to understand the fundamental behavior of metals and electronic phase transitions[32]. Several seminal ab initio works have reported the energy of HEG at different densities[21,22,32–35]. Recently two other works have extended neural network ansatz to study HEG[21,22]. Although our computational framework is independently designed for solids, the network structure between this work and refs. 21, 22 employ similar ideas. Different physics-inspired envelope functions and periodic

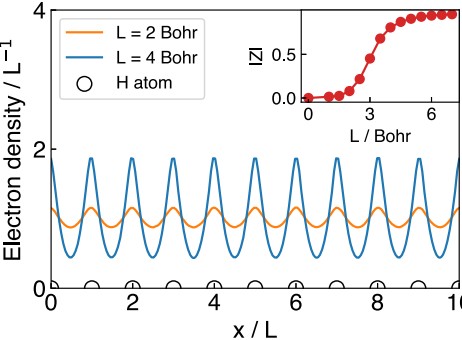

**Fig. 3 | Electron density of $H_{10}$ chains.** The horizontal axis is scaled by the corresponding bond length. Complex polarization modulus $|Z|$ as a function of bond length is plotted in the inset.

features are used in these works, which suit the features of solids and homogeneous electron gas respectively. We make comparisons between these networks and ours on HEG, and observe consistent performance, which further proves the effectiveness of neural network-based QMC works. In this section, we present the results calculated on a simple cubic cell containing 54 electrons in a closed-shell configuration, the largest HEG system studied in this work (Fig. 2g). More results and comparisons with other works on smaller systems are discussed in the section "Network comparison" and Supplementary Table 13.

Figure 2h shows our calculated correlation error on the 54-electrons HEG at six different densities from $r_s = 0.5$ Bohr to 20.0 Bohr. The state-of-the-art results, namely VMC with backflow correlation (BF)[33], distinguishable cluster with double excitations (DCD)[34], and transcorrelated full configuration interaction quantum Monte Carlo (TC-FCIQMC)[35] are also plotted for comparison, and BF-DMC result is often used as the reference energy of correlation error. Overall, our neural network performs very well, with an error of less than 1% in a wide range of density (see Supplementary Table 14). Generally, the correlation error increases as the density of HEG decreases when the correlation effects become larger, which also appears in DCD calculations.

## Electron density

Besides the total energy of solid systems, the electron density is also a key quantity to be calculated. For example, the electron density is crucial for characterizing different solids, such as the difference between insulators and conductors, and the distinct nature of ionic and covalent crystals. In DFT the one-to-one correspondence between many-body wavefunction and electron density proved by Hohenberg and Kohn in 1964 suggests that electron density is a fundamental quantity of materials. However, an interesting survey found that while new functionals in DFT improve the energy calculation the obtained density somehow can deviate from the exact[36]. Here, with our accurate neural network wavefunction, we can also obtain accurate electron density of solids and provide a valuable benchmark and guidance for method development.

A conductor features free-moving electrons, which would have macroscopic movements under external electric fields. In contrast, electrons are localized and constrained in insulators and cause considerable electron resistance. In Fig. 3, as an example, we show the calculated electron density of the hydrogen chain at two different bond lengths. As we can see, for the compressed hydrogen chain (L = 2 Bohr), the electron density is rather uniform and has small fluctuations. As the chain is stretched, the electrons are more localized and the density profile has larger variations. The observation is consistent with the well-known insulator-conductor transition on the hydrogen chain by varying the bond length. To measure the transition more

quantitatively, we further calculate the complex polarization $Z$ as the order parameter for insulator-conductor transition[37]. A conducting state is characterized by a vanishing complex polarization modulus $|Z|$ ~0, while an insulating state has a finite $|Z|$ ~1. We can see that the insulator-conductor transition bond length of the hydrogen chain is around 3 Bohr according to the calculated results, which is also consistent with the previous studies[37].

Ionic and covalent bonds are the most fundamental chemical bonds in solids. While the physical pictures of these two types of bonding are very different, they both lie in the behavior of electrons as the "quantum glue" and electron density distribution is a simple way to visualize different bonding types. Here we choose to calculate the electron density of the diamond-structured Si, rock-salt NaCl and LiH crystals at their equilibrium position. They are representative of covalent and ionic crystals, and have also been investigated by other high-level wavefunction methods, e.g., AFQMC[38]. Note that in the calculations of NaCl and Si, correlation-consistent effective core potential (ccECP) is employed to reduce the cost, which removes the inertia of core electrons and keeps the behavior of active valence electrons[15,39].

The electron density of diamond-structured Si in its (01$\bar{1}$) plane is plotted in Fig. 4b. We can see that valence electrons are shared by the nearest Si atoms, forming apparent Si-Si covalent bonds. In contrast, valence electrons are located around atoms in NaCl crystal as Fig. 4c shows. All the valence electrons are attracted around Cl atoms, forming effective $Na^+$ and $Cl^-$ ions in the crystal. Moreover, the electron density of LiH crystal is also calculated and plotted in Fig. 4d. LiH crystal is a moderate system between a typical ionic and covalent crystal. According to the result, the electrons are nearly equally distributed near Li and H atoms for our network. Detailed Bader charge analysis[40] manifests the atoms in the crystal become $Li^{0.67+}$ and $H^{0.67-}$ ions, respectively (resolution ~0.015 Å), which slightly deviates from the stable closed-shell configuration (see Supplementary Note 7 for more details).

## Network comparison

In refs. 21, 22, neural networks are also used to simulate homogeneous electron gas system, employing a different choice of periodic feature function. In Fig. 5 we plot the correlation error computed on the 14-electrons HEG system, which can be compared with the results of other works. We can see that all three networks can go beyond the BF-DMC level for high-density systems. For all systems tested, our correlation errors are about 2% with the TC-FCIQMC result as the reference[35], whereas the results of refs. 21, 22 are within 1%. It is understandable that the networks of refs. 21, 22 are specially designed for HEG systems, so slightly better accuracy can be achieved than our network. In their works, multiple phase factors $e^{ik\cdot r}$ are used in the constructed orbitals, which improve the expressiveness of the network. In comparison, our network contains an additional exponential decay term, which simulates the attraction between atoms and electrons in solids containing nuclei (see Methods section for more details). Furthermore, the choice of periodic distance, as well as the domains of the constructed wavefunction (complex or real-valued), are also different in these three works, which may add differences to their performance. In the future, it would be interesting to combine the insights learned from these three works and design a better network ansatz for periodic systems.

## Metallic lithium

We have also carried out preliminary calculations on metallic lithium. The real metal system remains a notoriously difficult task for accurate wavefunction approaches[7,41–44]. The zero gap of metal leads to a discontinuity in the Brillouin zone integral. As a consequence, a significantly larger simulation cell is required for metals than insulators to

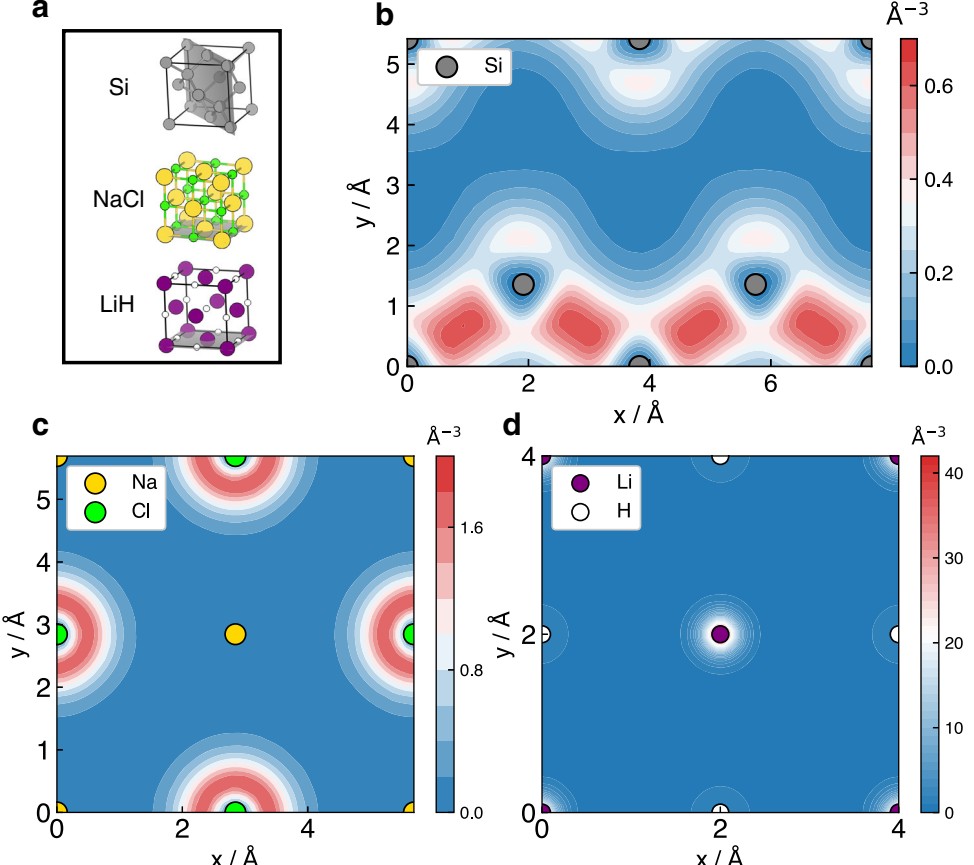

**Fig. 4 | Electron density of solids. a** Structures of solids, where the lattice planes for plotting electron densities are indicated. **b** Electron density of diamond-structured Si in its (01$\bar{1}$) plane, ccECP[Ne] is employed, and the bond length of Si equals 5.42 Å. **c** Electron density of NaCl crystal in its $xy$-plane, ccECP[Ne] is employed, and the bond length of NaCl equals 5.7 Å. **d,** the electron density of LiH crystal in its $xy$-plane, and the bond length of LiH equals 4.0 Å.

reach the thermodynamic limit. Shortcut approaches to simulate metals are proposed via employing a special twist angle[7,43], which helps to reduce the simulation size and finite-size error. Here we employ our network to simulate lithium with a body-centered cubic (bcc) structure, which is a typical metal with zero gap. A $2 \times 2 \times 2$ conventional cell of bcc-Li at $\Gamma$ point is employed (see Supplementary Table 11). In Supplementary Table 12, we list the total energy and the cohesive energy computed. As expected, the error in cohesive energy of lithium with such a limited supercell is larger than in non-metallic solids such as LiH, and further developments are desired to treat the large finite-size errors in metal.

## Discussion

The construction of a wavefunction for solid systems is a crucial but unsolved problem in the neural network community. The core mechanism of our neural network is the use of the periodic distance feature, which promotes molecule neural networks elegantly to the corresponding periodic ones and avoids time-consuming lattice summation. Considering the high-accuracy results obtained in this work, our neural network can be further applied to study more delicate physics and materials problems, such as the phase transitions of solids, surfaces, interfaces, and disordered systems, to name just a few. Our ansatz also offers a flexible extension to other neural networks and an easy integration into traditional computational techniques. The naturally evolved many-body wavefunction from the neural network may provide more physical and chemical insights into emergent phenomena of complex materials.

For further development of neural network-based QMC, the most crucial task is to enlarge its simulation size while retaining a reasonable accuracy, which allows a more accurate simulation of metals and high-temperature superconductors. Employing pseudopotential is helpful to enlarge the simulation size[15], while a better solution is a more efficient neural network, and related works are under progress.

## Methods

### Supercell approximation

Simulating a solid system requires solving the Schrödinger equation of many electrons within a large bulk. Supercell approximation is usually adopted to simplify the problem, considering a finite number of electrons and nuclei with periodic boundary conditions, whose Hamiltonian reads

$$\hat{H}_S = \sum_i -\frac{1}{2}\Delta_i + \frac{1}{2}\sum_{\mathbf{L}_S,i,j}' \frac{1}{|\mathbf{r}_i - \mathbf{r}_j + \mathbf{L}_S|}$$
$$- \sum_{\mathbf{L}_S,i,I} \frac{Z_I}{|\mathbf{r}_i - \mathbf{R}_I + \mathbf{L}_S|} + \frac{1}{2}\sum_{\mathbf{L}_S,I,J}' \frac{Z_I Z_J}{|\mathbf{R}_I - \mathbf{R}_J + \mathbf{L}_S|},$$

(3)

where $\mathbf{r}_i$ denotes the spatial position of $i$th electron in the supercell. $\mathbf{R}_I$, $Z_I$ are the spatial position and charge of the $I$th nucleus and $\{\mathbf{L}_S\}$ is the set of supercell lattice vectors, which is usually a subset of primitive cell lattice vectors $\{\mathbf{L}_p\}$. In order to simulate the real environments of electrons in solids, the interactions between the particles and their images are also included in $\hat{H}_S$, and the prime symbol in summation means $i = j$ terms are omitted for $\mathbf{L}_S = 0$.

Supercell Hamiltonian $\hat{H}_S$ is invariant under the translation of any electron by a vector in $\{\mathbf{L}_S\}$ as well as a simultaneous translation of

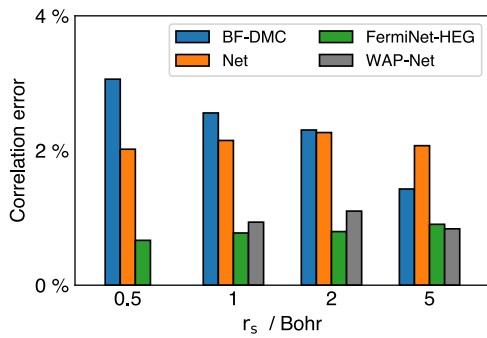

**Fig. 5 | Correlation error of 14-electrons HEG system at different $r_s$.** Correlation error is defined as $[1 - (E - E_{HF})/(E_{ref} - E_{HF})] \times 100\%$. WAP-Net refers to ref. 21 and FermiNet-HEG refers to ref. 22. BF-DMC results[33,34] are displayed for comparison, and TC-FCIQMC data were used as reference[35].

all-electrons by a vector in $\{\mathbf{L}_p\}$. As a consequence, two periodic conditions are required for the ground-state wavefunction $\psi$[45],

$$\psi(\mathbf{r}_1 + \mathbf{L}_p, \ldots, \mathbf{r}_N + \mathbf{L}_p) = \exp(i\mathbf{k}_p \cdot \mathbf{L}_p)\psi(\mathbf{r}_1, \ldots, \mathbf{r}_N),$$
$$\psi(\mathbf{r}_1 + \mathbf{L}_S, \ldots, \mathbf{r}_N) = \exp(i\mathbf{k}_S \cdot \mathbf{L}_S)\psi(\mathbf{r}_1, \ldots, \mathbf{r}_N), \tag{4}$$

where $\mathbf{k}_S$, $\mathbf{k}_p$ denote the momentum vectors reduced in the first Brillouin zone of the supercell and the primitive cell, respectively. Eq. (4) and the anti-symmetry condition together form the fundamental requirements for $\psi$. As the size of the supercell increases, simulation results gradually converge to the thermodynamic limit of a real solid system.

**Wavefunction ansatz**

In conventional QMC simulation of solids, Hartree–Fock type wavefunction ansatz composed of Bloch functions is often used, which reads

$$\psi^{HF}_{\mathbf{k}_S, \mathbf{k}_p}(\mathbf{r}) = \text{Det} \begin{vmatrix} e^{i\mathbf{k}_1 \cdot \mathbf{r}_1} u_{\mathbf{k}_1}(\mathbf{r}_1) & \cdots & e^{i\mathbf{k}_N \cdot \mathbf{r}_1} u_{\mathbf{k}_N}(\mathbf{r}_1) \\ \cdot & & \cdot \\ \cdot & & \cdot \\ \cdot & & \cdot \\ e^{i\mathbf{k}_1 \cdot \mathbf{r}_N} u_{\mathbf{k}_1}(\mathbf{r}_N) & \cdots & e^{i\mathbf{k}_N \cdot \mathbf{r}_N} u_{\mathbf{k}_N}(\mathbf{r}_N) \end{vmatrix}. \tag{5}$$

In order to satisfy Eq. (4), $\mathbf{k}_i$ in the determinant should lie on the grid of supercell reciprocal lattice vectors $\{\mathbf{G}_S\}$ offset by $\mathbf{k}_S$ within the first Brillouin zone of the primitive cell. Moreover, $u_{\mathbf{k}}$ functions in Eq. (5) should satisfy the translation invariant condition by the primitive cell lattice vectors,

$$u_{\mathbf{k}}(\mathbf{r} + \mathbf{L}_p) = u_{\mathbf{k}}(\mathbf{r}). \tag{6}$$

Following the strategy of FermiNet[10], Bloch functions in Eq. (5) can be promoted with collective distances,

$$e^{i\mathbf{k} \cdot \mathbf{r}_i} u_{\mathbf{k}}(\mathbf{r}_i) \rightarrow e^{i\mathbf{k} \cdot \mathbf{r}_i} u_{\mathbf{k}}(\mathbf{r}_i; \mathbf{r}_{\neq i}), \tag{7}$$

where $\mathbf{r}_{\neq i}$ denotes all the electron coordinates except $\mathbf{r}_i$. These collective orbitals are constructed to achieve the equivalence of electron permutations $P$,

$$P_{i,j} u_{\mathbf{k}_i}(\mathbf{r}_j; \mathbf{r}_{\neq j}) = u_{\mathbf{k}_j}(\mathbf{r}_i; \mathbf{r}_{\neq i}), \tag{8}$$

which combined with the Slater determinant ensures the antisymmetry nature of electrons. Moreover, we use the periodic distance features $d(\mathbf{r})$ in Eq. (2) to substitute ordinary $|\mathbf{r}|$ in the molecular neural

network. The periodic functions $f$, $g$ used in Eq. (2) read

$$f(\omega) = |\omega| \left(1 - \frac{|\omega/\pi|^3}{4}\right),$$
$$g(\omega) = \omega \left(1 - \frac{3}{2}|\omega/\pi| + \frac{1}{2}|\omega/\pi|^2\right), \tag{9}$$

and their arguments $\omega$ are reduced into $[-\pi, \pi]$. Eq. (6) can then be satisfied without causing discontinuity[20]. The constructed periodic features $\{\sum_i g(\omega_i)\mathbf{a}_i, d(\mathbf{r})\}$ are substituted into FermiNet[10] to build a periodic wavefunction. Specifically, electron-atom features $\mathbf{h}_e$ and electron–electron features $\mathbf{h}_{e,e'}$ are constructed as follows,

$$\mathbf{h}_e = \left\{ \Sigma_{i=1}^3 g(\omega_{e,I}^i)\, \mathbf{a}_i^p, d(\omega_{e,I}) \right\},$$
$$\mathbf{h}_{e,e'} = \left\{ \Sigma_{i=1}^3 g(\omega_{e,e'}^i)\, \mathbf{a}_i^S, d(\omega_{e,e'}) \right\}, \tag{10}$$

where $\omega_{e,I}$, $\omega_{e,e'}$ are defined as

$$\omega_{e,I} = (\mathbf{r}_e - \mathbf{R}_I) \cdot \left\{ \mathbf{b}_1^p, \mathbf{b}_2^p, \mathbf{b}_3^p \right\},$$
$$\omega_{e,e'} = (\mathbf{r}_e - \mathbf{r}_{e'}) \cdot \left\{ \mathbf{b}_1^S, \mathbf{b}_2^S, \mathbf{b}_3^S \right\}, \tag{11}$$

and superscripts $p$, $S$ denote the primitive cell and supercell respectively. A permutation equivalent feature $\mathbf{f}_e^\alpha$ are further constructed from $\mathbf{h}_e$, $\mathbf{h}_{e,e'}$,

$$\mathbf{f}_e^\alpha = \text{concat}(\mathbf{h}_e, \mathbf{g}^\uparrow, \mathbf{g}^\downarrow, \mathbf{g}_e^{\alpha,\uparrow}, \mathbf{g}_e^{\alpha,\downarrow}), \tag{12}$$

where $\alpha$ denotes the spin index ($\uparrow$, $\downarrow$). $\mathbf{g}^\uparrow$, $\mathbf{g}^\downarrow$ and $\mathbf{g}_e^{\alpha,\uparrow}$, $\mathbf{g}_e^{\alpha,\downarrow}$ read

$$(\mathbf{g}^\uparrow, \mathbf{g}^\downarrow) = \left( \frac{1}{n^\uparrow}\sum_e \mathbf{h}_e^\uparrow, \frac{1}{n^\downarrow}\sum_e \mathbf{h}_e^\downarrow \right),$$
$$(\mathbf{g}_e^{\alpha,\uparrow}, \mathbf{g}_e^{\alpha,\downarrow}) = \left( \frac{1}{n^\uparrow}\sum_{e'} \mathbf{h}_{e,e'}^{\alpha,\uparrow}, \frac{1}{n^\downarrow}\sum_{e'} \mathbf{h}_{e,e'}^{\alpha,\downarrow} \right). \tag{13}$$

$\mathbf{f}_e^\alpha$ and $\mathbf{h}_{e,e'}$ are subsequently substituted into a series of fully connected layers recursively

$$\mathbf{h}_e^{l+1,\alpha} = \tanh(\mathbf{V}^l \cdot \mathbf{f}_e^{l,\alpha} + \mathbf{b}^l) + \mathbf{h}_e^{l,\alpha},$$
$$\mathbf{h}_{e,e'}^{l+1,\alpha,\beta} = \tanh(\mathbf{W}^l \cdot \mathbf{h}_{e,e'}^{l,\alpha,\beta} + \mathbf{c}^l) + \mathbf{h}_{e,e'}^{l,\alpha,\beta}, \tag{14}$$

where $l$ denotes the number of layers, and $\{\mathbf{V}_l, \mathbf{b}_l\}$, $\{\mathbf{W}_l, \mathbf{c}_l\}$ denote corresponding weight and bias of $l$-layer.

Functions $u$ in Eq. (7) are built using the $\mathbf{h}_e^L$ from the last L-layer,

$$u = \text{Orb}^{Re} \cdot \mathbf{h}_e^L + \mathbf{i} \times \text{Orb}^{Im} \cdot \mathbf{h}_e^L, \tag{15}$$

where $\text{Orb}^{Re,Im}$ denote the weight parameters of the real part and the imaginary part respectively.

Moreover, $u$ function is multiplied by an additional phase factor $\exp(i\mathbf{k} \cdot \mathbf{r})$, which mimics Bloch functions and encodes the occupied $\mathbf{k}$-point information from HF calculation. Inspired by the tight-binding model in solid physics, a periodic-generalized envelope term $\exp[-d(\mathbf{r})]$ is also added to the molecule orbitals, which considers an attractive interaction effect between atoms and electrons. The final molecule orbitals $\phi$ reads

$$\phi(\mathbf{r}_i; \mathbf{r}_{\neq i}) = \exp(i\mathbf{k} \cdot \mathbf{r}_i) \exp[-d(\mathbf{r}_i)]u(\mathbf{r}_i; \mathbf{r}_{\neq i}). \tag{16}$$

For an overall sketch of the neural network, see section "Pseudocode of network". Note that the distance between electrons and nuclei is omitted for the HEG system since it does not contain any nucleus. Specific hyperparameters of each system are listed in Supplementary Note 1.

## Pseudocode of network

For clarity, the pseudocode of network reads below:

Require: electron positions $\{\mathbf{r}_1^\uparrow, \cdots, \mathbf{r}_{n^\uparrow}^\uparrow, \mathbf{r}_1^\downarrow, \cdots, \mathbf{r}_{n^\downarrow}^\downarrow\}$

Require: nuclear positions $\{\mathbf{R}_I\}$ in the primitive cell

Require: lattice vector $\{\mathbf{a}_1^{p,S}, \mathbf{a}_2^{p,S}, \mathbf{a}_3^{p,S}\}$ of primitive cell and supercell

Require: reciprocal lattice vector $\{\mathbf{b}_1^{p,S}, \mathbf{b}_2^{p,S}, \mathbf{b}_3^{p,S}\}$ of primitive cell and supercell

Require: occupied $\{\mathbf{k}_i\}$ points offered by Hartree–Fock method

For each electron $e$, atom $I$:

$\quad \omega_{e,I} = (\mathbf{r}_e - \mathbf{R}_I) \cdot \{\mathbf{b}_1^p, \mathbf{b}_2^p, \mathbf{b}_3^p\}$

$\quad \omega_{e,e'} = (\mathbf{r}_e - \mathbf{r}_{e'}) \cdot \{\mathbf{b}_1^S, \mathbf{b}_2^S, \mathbf{b}_3^S\}$

End For

For each electron $e$:

$\quad \mathbf{h}_e = \{\Sigma_{i=1}^3 g(\omega_{e,I}^i)\mathbf{a}_i^p, d(\omega_{e,I})\}$

$\quad \mathbf{h}_{e,e'} = \{\Sigma_{i=1}^3 g(\omega_{e,e'}^i)\mathbf{a}_i^S, d(\omega_{e,e'})\}$

End For

For each layer $l$:

$\quad \mathbf{g}^{l,\uparrow} = \frac{1}{n^\uparrow}\sum_e \mathbf{h}_e^{l,\uparrow}$

$\quad \mathbf{g}^{l,\downarrow} = \frac{1}{n^\downarrow}\sum_e \mathbf{h}_e^{l,\downarrow}$

$\quad$ For each electron $e$, spin $\alpha$:

$\quad\quad \mathbf{g}_e^{l,\alpha,\uparrow} = \frac{1}{n^\uparrow}\sum_{e'} \mathbf{h}_{e,e'}^{l,\alpha,\uparrow}$

$\quad\quad \mathbf{g}_e^{l,\alpha,\downarrow} = \frac{1}{n^\downarrow}\sum_{e'} \mathbf{h}_{e,e'}^{l,\alpha,\downarrow}$

$\quad\quad \mathbf{f}_e^{l,\alpha} = \mathrm{concat}(\mathbf{h}_e^{l,\alpha}, \mathbf{g}^{l,\uparrow}, \mathbf{g}^{l,\downarrow}, \mathbf{g}_e^{l,\alpha,\uparrow}, \mathbf{g}_e^{l,\alpha,\downarrow})$

$\quad\quad \mathbf{h}_e^{l+1,\alpha} = \tanh(\mathbf{V}^l \cdot \mathbf{f}_e^{l,\alpha} + \mathbf{b}^l) + \mathbf{h}_e^{l,\alpha}$

$\quad\quad \mathbf{h}_{e,e'}^{l+1,\alpha,\beta} = \tanh(\mathbf{W}^l \cdot \mathbf{h}_{e,e'}^{l,\alpha,\beta} + \mathbf{c}^l) + \mathbf{h}_{e,e'}^{l,\alpha,\beta}$

$\quad$ End For

End For

For each orbital $i$:

$\quad$ For each electron $e$, spin $\alpha$:

$\quad\quad u_{i,e}^\alpha = \mathrm{Orb}_{i,\alpha}^{\mathrm{Re}} \cdot \mathbf{h}_e^L + \mathbf{i} \times \mathrm{Orb}_{i,\alpha}^{\mathrm{Im}} \cdot \mathbf{h}_e^L$

$\quad\quad p_{i,e}^\alpha = \exp(\mathbf{i}\mathbf{k}_i \cdot \mathbf{r}_e^\alpha)$

$\quad\quad \mathrm{enve}_{i,e}^\alpha = \sum_I \pi_i^{I,\alpha} \exp[-\sigma_i^{I,\alpha} d(\omega_{e,I})]$

$\quad\quad \phi_{i,e}^\alpha = p_{i,e}^\alpha u_{i,e}^\alpha \mathrm{enve}_{i,e}^\alpha$

$\quad$ End For

End For

$\psi = \mathrm{Det}[\phi^\uparrow]\mathrm{Det}[\phi^\downarrow]$

## Neural network optimization

Parameters $\theta$ within the neural network can be optimized to minimize the energy expectation value $\langle E_l \rangle$, and the gradient $\nabla_\theta \langle E_l \rangle$ reads

$$\nabla_\theta \langle E_l \rangle = \mathrm{Re}[\langle E_l \nabla_\theta \ln\psi^* \rangle - \langle E_l \rangle \langle \nabla_\theta \ln\psi^* \rangle],$$
$$E_l = \psi^{-1}\hat{H}_S\psi, \quad (17)$$

where $E_l$ denotes the local energy of neural network ansatz $\psi$. Besides energy minimization, stochastic reconfiguration optimization[46] has also been widely adopted and proved to be much more efficient, whose gradient reads

$$\mathrm{Grad} = F^{-1}\nabla_\theta \langle E_l \rangle,$$
$$F_{ij} = \mathrm{Re}\left[\left\langle \frac{\partial \ln\psi^*}{\partial \theta_i}\frac{\partial \ln\psi}{\partial \theta_j}\right\rangle - \left\langle \frac{\partial \ln\psi^*}{\partial \theta_i}\right\rangle\left\langle \frac{\partial \ln\psi}{\partial \theta_j}\right\rangle\right]. \quad (18)$$

In this work, we adopt a modified KFAC optimizer, which approximates $F$ as

$$F = \mathrm{Re}\left[\left\langle \frac{\partial \ln\psi^*}{\partial \mathrm{vec}(W_l)}\frac{\partial \ln\psi^T}{\partial \mathrm{vec}(W_l)}\right\rangle - \left\langle \frac{\partial \ln\psi^*}{\partial \mathrm{vec}(W_l)}\right\rangle\left\langle \frac{\partial \ln\psi^T}{\partial \mathrm{vec}(W_l)}\right\rangle\right]$$
$$= \mathrm{Re}\left[\langle (a_l \otimes e_l^*)(a_l \otimes e_l)^T \rangle - \langle (a_l \otimes e_l^*)\rangle\langle (a_l \otimes e_l)\rangle^T\right] \quad (19)$$
$$\approx \mathrm{Re}\left[\langle a_l a_l^T\rangle \otimes \langle e_l^* e_l^T\rangle\right],$$

where $W_l$ denotes the weight parameters of layer $l$, and vec means vectorized form. $a_l$, $e_l$ denote the activation and sensitivity of layer $l$ respectively. Note that activation $a_l$ is always real-valued, which explains the absence of conjugation of $a_l$ in the second line. The first term in the bracket of Eq. (19) is approximated as the Kronecker product of the expectation values, and the second term is omitted for simplification.

## Twist average boundary condition

TABC is a conventional technique to reduce the finite-size error due to the constrained size of the supercell[27]. It averages the contributions from different periodic images of the supercell and improves the convergence of the total energy. The formula reads

$$E_{\mathrm{TABC}} = \frac{\Omega_S}{(2\pi)^3}\int_{1.\mathrm{B.Z.}} d^3\mathbf{k}_S \frac{\psi_{\mathbf{k}_S}^*\hat{H}_S\psi_{\mathbf{k}_S}}{\psi_{\mathbf{k}_S}^*\psi_{\mathbf{k}_S}}, \quad (20)$$

where 1.B.Z. denotes the first Brillouin zone of supercell and the integral is practically approximated by a discrete sum of a Monkhorst-Pack mesh (see Supplementary Note 3.2 for more details).

## Structure factor correction

Finite-size error can be further reduced via the structure factor $S(\mathbf{k})$ correction[28], which is usually calculated to correct the exchange-correlation potential $V_{\mathrm{xc}}$ and the formula reads

$$\frac{\Delta V_{\mathrm{xc}}}{N_e} = \frac{2\pi}{\Omega_S}\lim_{\mathbf{k}\to 0}\frac{S(\mathbf{k})}{\mathbf{k}^2},$$
$$S(\mathbf{k}) = \frac{1}{N_e}\left[\langle\rho(\mathbf{k})\rho^*(\mathbf{k})\rangle - \langle\rho(\mathbf{k})\rangle\langle\rho^*(\mathbf{k})\rangle\right], \quad (21)$$

where $\lim_{\mathbf{k}\to 0}$ is practically estimated via interpolation (see Supplementary Note 3.4 for more details).

## Empirical correction formula

Empirical formulas are also commonly employed to reduce the finite-size error[18], one of which reads

$$E_\infty = E_{\mathrm{N}}^{\mathrm{Net}} + \left(E_\infty^{\mathrm{HF}} - E_{\mathrm{N}}^{\mathrm{HF}}\right). \quad (22)$$

The simulation size of high-accuracy methods is usually limited due to high computational costs. Hence methods with a much more practical time scale, such as HF, is usually used to give a posterior estimation of the finite-size error. All the results of LiH are corrected using this empirical formula with HF results in a cc-pVDZ basis (see Supplementary Note 4.3 for more details).

## Electron density analysis

Electron density $\rho(\mathbf{r})$ is defined as

$$\rho(\mathbf{r}) = N\int d^3\mathbf{r}_2\cdots d^3\mathbf{r}_N|\psi(\mathbf{r},\mathbf{r}_2,\cdots,\mathbf{r}_N)|^2, \quad (23)$$

and it's practically evaluated by accumulating Monte Carlo samples of electrons on a uniform grid over the simulation cell. As for the complex polarization $Z$, it is defined as[37]

$$Z = \left\langle \exp\left(i\sum_i \frac{2\pi}{L}\mathbf{r}_i^\|\right)\right\rangle, \quad (24)$$

where $\mathbf{r}^\|$ denotes the projection of electron coordinate along the chain direction. Moreover, Bader charge is employed to estimate the charge partition on each atom[40]. The convergence test of Bader charge is shown in the Supplementary Fig. 8.

### Workflow and computational details

This work is developed upon open-source FermiNet[47] and PyQMC[48] on Github, deep learning framework JAX[49] is used which supports flexible and powerful complex number calculation. Ground-state energy calculations are performed with all-electrons. Diamond-structured Si and NaCl crystal are simulated with ccECP[Ne][39]. The neural network is pretrained by Hartree−Fock ansatz, obtained with PySCF software[50]. All the used **k** points are the occupied **k** points from Hartree−Fock calculation using Monkhorst-Pack mesh offset by $\mathbf{k}_S$ in cc-pVDZ basis, and the mesh size is the same as the supercell. All the expectation values for distribution $|\psi|^2$ are evaluated via the Monte Carlo approach, and then the energy and wavefunction is optimized using the modified KFAC optimizer[24] (see Supplementary Figs. 1, 2, 4, 6, 7). The Ewald summation technique is implemented for the lattice summation in energy calculation. After training is converged, energy is calculated in a separate inference phase.

### Reporting summary

Further information on research design is available in the Nature Portfolio Reporting Summary linked to this article.

## Data availability

The data generated in this study are provided in the Supplementary Information.

## Code availability

The concrete code of this work is developed on Github (https://github.com/bytedance/DeepSolid).

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

## Acknowledgements

The authors thank Matthew Foulkes, David Ceperley, Lucas Wagner, Gareth Conduit, Mario Motta, and Ke Liao for helpful discussions. We thank Gino Cassella for providing Hartree–Fock energies of HEG. We thank the ByteDance AML team specially for their technical and computing support. We also thank ByteDance AI-Lab LIT Group and the rest of the ByteDance AI-Lab research team for inspiration and encouragement. This work is directed and supported by Hang Li and ByteDance AI-Lab. J.C. is supported by the National Natural Science Foundation of China under Grant No. 92165101.

## Author contributions

X.L. and J.C. conceived the study; X.L. developed the method, performed implementations, simulations, and data analyses; Z.L. contributed to the code development and simulation of HEG; J.C. supervised the project. X.L., Z.L., and J.C. wrote the paper.

## Competing interests

The authors declare no competing interests.
