## [Peer Review File · Nature Communications]

Ab initio calculation of real solids via neural network ansatzREVIEWER COMMENTS

Reviewer #1 (Remarks to the Author):

The paper presents an extension of machine learning-based algorithms to solving the quantum-many body problem with periodic boundary condition, as well as results of the study of the thermodynamic limit of various systems. The construction of the algorithm is not as novel as one would have expected, but the problem treated is very important, and the results are quite carefully done and presented. The paper is well written and easy to read.

I am not quite sure about the standards of the journal. If the paper was submitted to Physical Review Letters,
I would recommend accepting the paper as it is.

Reviewer #2 (Remarks to the Author):

In this paper, Xiang Li et al. extended FermiNet by David Pfau et al. to periodic systems using Whitehead et al.'s periodic distance feature. They achieved high-precision quantum chemical calculations of real solids such as LiH crystals. This work is beautiful. I think this is an important work that will lead to the further development of neural network quantum states, which have been actively researched in recent years. The fact that the research code has been made publicly available is also excellent. This paper definitely deserves to be published in Nature Comm. but improvements are mandatory prior to its publication. Comments are provided below.

1. Applying neural network for ab initio calculations of real solids has been already proposed by another group (<https://doi.org/10.1038/s42005-021-00609-0>). The research group has extended the restricted Boltzmann Machine ansatz to real periodic systems, although it employs the second quantization and finite basis approximation. The authors should mention this work in the introduction.
2. The authors states that the coupled cluster theory 'face severe difficulty when applied to solid systems.' Does this simply mean that the pre-factor of computational cost is high, or does it essentially mean that there is something wrong with its application to solids? Please clarify this point.
3. To my knowledge, Bloch functions are for one-electron states not for correlated many-body wavefunctions. Besides, the FermiNet employs the Generalized Slater determinants and its 'molecular orbitals' actually take into account electron correlations. The authors should prove that the application of the Bloch function form is not an approximation(if it is an approximation, please explicitly mention that). Unless this is a very trivial matter, the authors should state the proof in the Appendix or Supplementary Information.
4. KFAC and the mechanism of using mutiple GPUs efficiently were introduced to FermiNet by the DeepMind team not by the authors. Please write so that this is clear to the reader.
5. There are no information on the k-points employed. The authors should describe how many k-points were used and how to generate the k-point grids.
6. The definition of "Correlation error" in the Fig. 2 h is not given.
7. 'ccpvdz' is not the correct spelling. cc-pVDZ is correct and should be corrected.
9. Please position the figure captions above each table in the Supplementary Information.
10. I have never seen the word 'TZ-LDA basis.' I googled it but cannot found. The authors should write so that it is clear for readers what basis functions were used for the reproducibility of the calculation.

11. Please specify actual computational costs of this method. For example, the wall time taken to calculate LiH and the computational resources used.
12. Supplementary Table 4 looks listing energies per atom. Please state this clearly.
13. All the statistical errors written in the Supplementary Information are '(1)' except Supplementary Tables 6, 10 and 11. Is this a coincidence?
14. There is no mention to the statistical error in the main text. For example, cohesive energy per unit cell is listed to three decimal places in eV, but without a description of the statistical error. Is the statistical error that small?
15. My understanding is that as long as finite k-point sampling is used, the divergent term derived from the exchange integral should appear at the Γ point. Is there no need for any correction to this divergent term? (c.f. DOI:10.1063/1.1926272, DOI:0.1021/acs.jctc.7b00049)

Reviewer #3 (Remarks to the Author):

Below, I review the strengths and weaknesses of this manuscript before providing a recommendation.

The strengths of the manuscript are as follows:

1. The authors have found a way to connect FermiNet (Ref 10, 11, 31) to another open-source code, PySCF (Ref 39), and this allows for the general treatment of solid state systems. Along the way, they propose a different way to manage periodic boundary conditions compared to the authors of FermiNet, who recently showed that their method can be used on the Wigner crystal (Ref 31), an inhomogeneous system with periodic boundary conditions. The periodic boundary condition considerations are based on Refs. 18.
2. It is commendable, from a methods development standpoint, that a good number of examples are provided with a variety of benchmarks that do show that this is a general approach. The systems shown do have a broad coverage, especially considering that this is first implementation - covering a model (UEG), insulator, semi metal sheet, and 1D chain. The comparisons with experiment are also positive.
3. The manuscript is well written and this is a topical area. Any resultant publication will be well cited.

The weaknesses of this manuscript are as follows:

4. The manuscript rests heavily on connecting software and techniques from prior authors (Refs. 10, 11, 18, 26, 31, 39); the specific originality of this manuscript is not clearly articulated.
5. The manuscript's claim that the method outperforms other ab initio methods is not supported by the evidence shown. For the H10 and the UEG, the result is better stated as similar performance with the state-of-the-art DMC results (rather than a claim of being better than common approaches). In the case of the UEG in particular, it is known that the DMC results are not free from error at high densities due to fixed node error. Instead, as highlighted by a recent paper (Ref 31), the most accurate approaches come from FCIQMC which are available for $N=14$. Similar FCIQMC results are also available for $N=54$, which could have been used for comparison as they show clear improvement on BF-DMC.
6. The manuscript's claim of improvement over the existing FermiNet papers is not supported by the evidence shown. The changes made include a different choice of PBCs and NN optimizer. To validate the effectiveness of the optimizer, we would need to see efficiency data. For the PBCs choice, we would need to see well-analyzed side by side comparisons. Instead, the manuscript does not reach a direct comparison when treating the UEG, choosing different systems than Ref

31. It is possible to infer from the data presented that this algorithm captures DMC-quality energies, while Ref 31 goes beyond DMC to FCIQMC level accuracy. Thus, this manuscript's choice of PBCs seems less accurate than FermiNet not more.

7. The manuscript's claim of investigating long-range effectiveness in real systems is weakened by the size and electronic structure of the systems chosen. In practice, the long-range components of the energy are being added after the fact with finite size corrections (Ref. 26). The calculations themselves are being performed on relatively small system sizes. Evidence is not presented for the convergence of long-range effects (which would, for example, require analysis of the energy with system size). Moreover, the systems shown do not include a real metal (where long-range effects really come into play) and the system sizes shown have significant gaps at the one-particle/HF level of theory. So, while the systems proposed are ab initio real solids, they are not actually treated in the same way as the state of the art, and do not go beyond what can already be done.

Assessment of comparable works/established literature

Comparing this paper to what is currently the state of the art in the field, there are a number of high impact publications. The best known paper is by Troyer in Science (Ref 9) and the original FermiNet paper was published in Phys Rev Research (Ref 10). The best case scenario of the technical development appears to be a more efficient version of FermiNet for solids and I would put this on a par with the original FermiNet development in terms of its specialism. In my view, this paper is better suited to a more specialized journal which would also allow the claims of the novelty of the network and any speed up to be better considered and reviewed.

Summary

I think this is an interesting paper and it is important that it is published. However, my recommendation would be to seek publication in a journal that is well aligned with methods development.

If the applications are able to be significantly improved, this may find home in a more specialized Nature publishing group journal related to computation or materials science. I would be supportive of this paper being transferred to NPJ Computational Materials. While I do not think it would be suitable for publication in its current work at NPJ Computational Materials, I would be willing/able to provide further comments that -- if acted upon -- would make it suitable. This would require the authors to show side by side comparisons with FermiNet (Ref 31) that show clear improvement. I also think the authors could collect more timing data and show improvement in that way, then they might reasonably try to submit to Nature Computational Science.

On the other hand, I think it would be immediately publishable in a journal such as the Journal of Chemical Theory and Computation, Phys Rev B, or Phys Rev Research.

Response to reviewers

Reviewer #1

The paper presents an extension of machine learning-based algorithms to solving the quantum-many body problem with periodic boundary condition, as well as results of the study of the thermodynamic limit of various systems. The construction of the algorithm is not as novel as one would have expected, but the problem treated is very important, and the results are quite carefully done and presented. The paper is well written and easy to read.

I am not quite sure about the standards of the journal. If the paper was submitted to Physical Review Letters,

I would recommend accepting the paper as it is.

We thank reviewer #1 for his/her time and the positive assessments of the importance of our work.

Reviewer #2

In this paper, Xiang Li et al. extended FermiNet by David Pfau et al. to periodic systems using Whitehead et al.'s periodic distance feature. They achieved high-precision quantum chemical calculations of real solids such as LiH crystals. This work is beautiful. I think this is an important work that will lead to the further development of neural network quantum states, which have been actively researched in recent years. The fact that the research code has been made publicly available is also excellent. This paper definitely deserves to be published in Nature Comm. but improvements are mandatory prior to its publication. Comments are provided below.

We thank reviewer #2 for his/her enthusiasm and positive comments.

Below are our point-to-point response to reviewer #2.

1. Applying neural network for ab initio calculations of real solids has been already proposed by another group (<https://doi.org/10.1038/s42005-021-00609-0>). The research group has extended the restricted Boltzmann Machine ansatz to real periodic systems, although it employs the second quantization and finite basis approximation. The authors should mention this work in the introduction.

Thanks for supplying related works. They are now properly cited and mentioned in our revision.

2. The authors states that the coupled cluster theory 'face severe difficulty when applied to solid systems.' Does this simply mean that the pre-factor of computational cost is high, or does it essentially mean that there is something wrong with its application to solids? Please clarify this point.

We think CC theory has quite high computational scaling and that's what we mean '**difficulties**'. And we have clarified it in our revision.

3. To my knowledge, Bloch functions are for one-electron states not for correlated many-body wavefunctions. Besides, the FermiNet employs the Generalized Slater determinants and its 'molecular orbitals' actually take into account electron correlations. The authors should prove that the application of the Bloch function form is not an approximation(if it is an approximation, please explicitly mention that). Unless this is a very trivial matter, the authors should state the proof in the Appendix or Supplementary Information.

In our work, we actually employed a **generalized many-body** Bloch function $u_{\mathbf{k}}(\mathbf{r}_i; \mathbf{r}_{\neq i})$, which is similar to the collective molecular orbitals in FermiNet. And its form reads

$$u_{\mathbf{k}}(\mathbf{r}_i; \mathbf{r}_{\neq i}) = \exp(i\mathbf{k} \cdot \mathbf{r}_i)\phi_{\mathbf{k}}(\mathbf{r}_i; \mathbf{r}_{\neq i})$$

where $\exp(i\mathbf{k} \cdot \mathbf{r}_i)$ is one-body electron phase factor, and $\phi_{\mathbf{k}}(\mathbf{r}_i; \mathbf{r}_{\neq i})$ are **collective** molecular orbitals from ferminet which take into account electron correlations.

The generalized many-body Bloch function does not introduce approximations such as single-electron approximation. The ansatz based on the generalized many-body Bloch function is a valid many-body wave function for solids. In this framework, as long as the collective molecular orbitals are sufficiently expressed by neural networks, our many-body wave function can approach the true ground state. This is added to the revised manuscript.

4. KFAC and the mechanism of using mutiple GPUs efficiently were introduced to FermiNet by the DeepMind team not by the authors. Please write so that this is clear to the reader.

Thanks for mentioning this and we have given deserved credits to DeepMind in our revision.

5. There are no information on the k-points employed. The authors should describe how many k-points were used and how to generate the k-point grids.

Actually, k-points in our works are all the occupied k-points from HF calculations in cc-pVDZ basis using MP mesh, and the mesh size is the same as the supercell. We mentioned it explicitly in the "Workflow and computational details" section now.

6. The definition of "Correlation error" in the Fig. 2 h is not given.

The definition of "Correlation error" is defined as

$$\text{Correlation error} = \left(1 - \frac{E - E_{\text{HF}}}{E_{\text{ref}} - E_{\text{HF}}}\right) \times 100\%$$

Where E_{HF} and E_{ref} refer to corresponding Hartree-Fock energy and reference energy (BF-DMC in our work for HEG) respectively. Its definition has been added in the caption of Fig. 2 in our revision.

7. 'ccpvdz' is not the correct spelling. cc-pVDZ is correct and should be corrected.

Thanks for pointing this out and the spellings are corrected now.

8. Please position the figure captions above each table in the Supplementary Information.

The figure captions have been properly positioned now in our revision.

9. I have never seen the word 'TZ-LDA basis.' I googled it but can not found. The authors should write so that it is clear for readers what basis functions were used for the reproducibility of the calculation.

Sorry for the misleading wording. We have checked the original paper "*Towards the Solution of the Many-Electron Problem in Real Materials: Equation of State of the Hydrogen Chain with State-of-the-Art Many-Body Methods*". (see appendix A.2.b of it.)

Cited VMC and DMC results are actually calculated using cc-pVTZ basis. And their one-body Jastrow function includes orbitals from LDA calculation. That's what 'TZ-LDA' basis means. We have now added the corrected details in the caption of Fig. 2.

10. Please specify actual computational costs of this method. For example, the wall time taken to calculate LiH and the computational resources used.

The used computational resources and wall time are listed in Supplementary Table 2 now.

11. Supplementary Table 4 looks listing energies per atom. Please state this clearly.

You are right and we have stated it explicitly in our revision. Necessary explanations are also added in other captions.

12. All the statistical errors written in the Supplementary Information are '(1)' except Supplementary Tables 6, 10 and 11. Is this a coincidence?

We have double checked our data, as well as the reference, and we think it's a coincidence. The listed energy results are calculated using enough inference steps ($5e4$) and our trained ansatz is close to the eigen-state. We think that's why the standard error is so small.

13. There is no mention to the statistical error in the main text. For example, cohesive energy per unit cell is listed to three decimal places in eV, but without a description of the statistical error. Is the statistical error that small?

This question is related to point 12 and we have double checked the data. The statistical error of our calculated LiH crystal is $1e-5$ Ha/cell which leads to nearly $3e-4$ eV/cell error to cohesive energy. It's quite small compared to the calculated -4.7 eV/cell results and we choose to omit it. Our trained ansatz is close to the eigen-state. We think that's why the statistical error is so small. We have added a statement in the caption of Fig.2 to clarify this.

14. My understanding is that as long as finite k-point sampling is used, the divergent term derived from the exchange integral should appear at the Γ point. Is there no need for any correction to this divergent term? (c.f. DOI:10.1063/1.1926272, DOI:0.1021/acs.jctc.7b00049)

You are totally right. At QMC level, we also need to correct the exchange-correlation energy due to integral error, and structure factor correction is usually used. In our work, we follow the standard procedures (arXiv:cond-mat/0605004, arXiv:0806.0957v2) to reduce the integral error for graphene. This is detailed in the Method section.

Reviewer #3

The strengths of our work mentioned by reviewer #3 are listed below:

1. The authors have found a way to connect FermiNet (Ref 10, 11, 31) to another open-source code, PySCF (Ref 39), and this allows for the general treatment of solid state systems. Along the way, they propose a different way to manage periodic boundary conditions compared to the authors of FermiNet, who recently showed that their method can be used on the Wigner crystal (Ref 31), an inhomogeneous system with periodic boundary conditions. The periodic boundary condition considerations are based on Refs. 18.

2. It is commendable, from a methods development standpoint, that a good number of examples are provided with a variety of benchmarks that do show that this is a general approach. The systems shown do have a broad coverage, especially considering that this is first implementation -- covering a model (UEG), insulator, semi metal sheet, and 1D chain. The comparisons with experiment are also positive.
3. The manuscript is well written and this is a topical area. Any resultant publication will be well cited.

We thank reviewer #3 for acknowledging the strengths and potential impacts of our work, which is carried out **independently and timely**.

The weaknesses of our work mentioned by reviewer #3 are listed below, and we will respond point-by-point.

1. The manuscript rests heavily on connecting software and techniques from prior authors (Refs. 10, 11, 18, 26, 31, 39); the specific originality of this manuscript is not clearly articulated.

The originality of our work is quite clear. We **independently** propose a **new** neural network ansatz for solid systems, which allows us to carry out the first study of real solids using neural network in the framework of first quantization.

Our work is mainly inspired by molecular neural network (FermiNet) (Refs. 10,11) and periodic features designed for treating solids with traditional quantum Monte Carlo (Ref. 18, renumbered as ref. 20 in the revision). Although neural networks have shown success for molecules, their powerfulness have not been examined for real solids. Therefore, our work is definitely original and timely. Thanks to the reviewer's comment, we have made modifications to further clarify the motivation of this work, relevant research and references.

Ref. 31 (Ref. 34 in the revision) is a recent work extending FermiNet to study periodic model (homogeneous electron gas), which overlaps with the electron gas section of our work. We would like to point out that our study is **independent**. At the time Ref. 31 were released on arXiv in February 2022, we have already completed our code development and also most of our experiments. We have clarified this in the revision.

Ref. 26 (Ref. 29 in the revision) reports a commonly-used method to reduce the finite-size error in QMC. Correcting finite-size error is a standard task in nearly all the QMC calculations of solids, and so is in our QMC work. Developing a better algorithm for such purpose is a different topic concerned in QMC community, and is not the aim of this work. So we do not agree that adopting such a method is a weakness of this work.

As for Ref. 39 (Ref. 43 in the revision), it is a reference for the self-consistent field software PySCF, which is a popular and powerful tool to provide starting point for many-body wave function calculations. All QMC softwares (e.g. CASINO, QMCPACK, PyQMC) rely on external self-consistent field software to give an initial wave function, and so does our QMC work or other neural networks (FermiNet, PauliNet). We use Hartree Fock wave function calculated from PySCF to pretrain our network. And actually, our simulation quality doesn't depend much on pretrain after KFAC optimizer is adopted. Again, we don't think integrating our method with PySCF should be considered as a weakness.

2. The manuscript's claim that the method outperforms other ab initio methods is not supported by the evidence shown. For the H10 and the UEG, the result is better stated as similar performance with the state-of-the-art DMC results (rather than a claim of being better than common approaches). In the case of the UEG in particular, it is known that the DMC results are not free from error at high densities due to fixed node error. Instead, as highlighted by a recent paper (Ref 31), the most accurate approaches come from FCIQMC which are available for $N=14$. Similar FCIQMC results are also available for $N=54$, which could have been used for comparison as they show clear improvement on BF-DMC.

We have double-checked our manuscript and it is not obvious to us which statement about the performance of our method is unsupported.

We find the following statements in our manuscript where we make performance claims.

For Hydrogen chain:

We can see that our results nearly coincide with the LR-DMC results and significantly **outperform** traditional VMC (see Supplementary Table 3). (from our original manuscript, line 109)

Our TDL result is **comparable** with both AFQMC and LR-DMC. (from our original manuscript, line 116)

We think it's quite obvious from Fig.2a and 2b (see also below) that our result is much better than traditional VMC since our calculated energy is much lower (the lower the better).

Moreover, our results coincide with AFQMC and LR-DMC, so we claim "**comparable**".

For H(U)EG:

Overall, our neural network **performs very well**, with an error of less than 1% in a wide range of density (see Supplementary Table 11). (from our original manuscript, line 176)

We think it's also evident from Fig.2h (see also below) that our network performs very well. It has lower correlation error than BF-VMC at high densities ($r_s = 0.5, 1.0, 2.0$), and DCD at low densities ($r_s = 5, 10, 20$). We think it's reasonable for us to claim that it **"performs very well"**.

Moreover, thanks for informing us of the DMC problems at high density for HEG. We reworded the description of DMC accuracy level in HEG as "... and the most accurate BF-DMC result is **often** used as the reference energy of correlation error".

We think these claims made are reasonable and supported by our data. We will be grateful if the reviewers can point out explicitly where our statements are inappropriate, and we are happy to make further revisions.

3. The manuscript's claim of improvement over the existing FermiNet papers is not supported by the evidence shown. The changes made include a different choice of PBCs and NN optimizer. To validate the effectiveness of the optimizer, we would need to see efficiency data. For the PBCs choice, we would need to see well-analyzed side by side comparisons. Instead, the manuscript does not reach a direct comparison when treating the UEG, choosing different systems than Ref 31. It is possible to infer from the data presented that this algorithm captures DMC-quality energies, while Ref 31 goes beyond DMC to FCIQMC level accuracy. Thus, this manuscript's choice of PBCs seems less accurate than FermiNet not more.

We never claimed that we have made any improvement to the existing FermiNet papers in our manuscript. We are not sure which sentence of our manuscript leads to the reviewer's this concern. Instead, it is clearly presented that our work is an extension of the molecular neural network FermiNet to solids by implementing periodic features. Calculation of real solids has not been conducted in previous FermiNet papers, and is not possible without methodological developments reported in this work.

The reviewer suggested that we compare our results with Ref. 31 on the HEG system. We followed this suggestion and performed additional simulations. Below are our results.

Energy per particle of closed-shell HEG (14 electrons, cubic, Hartree)				
r_s	Our network	FermiNet-HEG (Ref.31)	BF-DMC(a)	TC-FCIQMC(a)
0.5	3.413258(4)	3.412683(4)	3.41370(2)	3.41241(1)
1	0.569425(2)	0.568904(2)	0.56958(1)	0.56861(1)
2	-0.007961(1)	-0.0084275(7)	-0.007949(7)	-0.00868(2)
5	-0.0795646(3)	-0.0798213(7)	-0.079706(3)	-0.08002(2)

a. Ke Liao et. al. Towards efficient and accurate ab initio solutions to periodic systems via transcorrelation and coupled cluster theory. Phys. Rev. Research 3, 033072 (2021)

We can see our network goes beyond DMC and approaches FCIQMC level at $r_s = 0.5, 1, 2$, which is similar to Ref.31. And Ref.31 reaches an **even better** accuracy than us, which is **understandable since** Ref.31 is specifically designed for HEG while our network is designed for real solids. Many parts, such as the envelope functions, are discarded when we simulate HEG.

The comparison between us and Ref.31 is discussed more in our revision, and the additional data is included in the revised supplementary information.

- The manuscript's claim of investigating long-range effectiveness in real systems is weakened by the size and electronic structure of the systems chosen. In practice, the long-range components of the energy are being added after the fact with finite size corrections (Ref. 26). The calculations themselves are being performed on relatively small system sizes. Evidence is not presented for the convergence of long-range effects (which would, for example, require analysis of the energy with system size). Moreover, the systems shown do not include a real metal (where long-range effects really come into play) and the system sizes shown have significant gaps at the one-particle/HF level of theory. So, while the systems proposed are ab initio real solids, they are not actually treated in the same way as the state of the art, and do not go beyond what can already be done.

We admit that our simulation size is relatively small compared to traditional QMC. It is always difficult to improve the accuracy and efficiency simultaneously when developing a new method.

In this work, it is quite clear that the main point is to extend the application of neural networks to solids, where the main motivation comes from the fact that neural networks have demonstrated better accuracy than traditional methods for molecules. Our work shows neural networks are also promising for solids. In addition, to the best of our knowledge, the simulation of the $3 \times 3 \times 3$ supercell of LiH (108 electrons) reported in this work is currently the largest system reported in the neural network community. Such a size is already very impressive and exceeds the limit of some traditional high-order quantum chemistry methods.

In solids, indeed there are other problems to overcome, such as the dealing of finite-size effects. However, finite-size correction methods are clearly quite different topics and should be focused on in separate works. We want to clarify that we **never claimed** that the finite-size error does not occur in our calculations. In this work, we employ the established methods in the community, such as TABC and structure factor correction in Refs.25 and 26. These corrections are also employed widely in traditional (state-of-the-art) QMC works. We also did not claim we go beyond what can already be done by traditional methods of the state-of-the-art. Our results are promising because we can reach comparable results to the state-of-the-art calculations and experimental measurements for solids.

Furthermore, the reviewers suggested that we study metal where finite-size error is more significant. Large finite-size error in metal are a well-known problem in the community. As we said, a discussion of finite size errors in metal would be a very different topic, see for example a recent work by Mihm et al. (Nature Comput. Sci. 1, 801, 2021). In this work, we focus on the applicability of our neural networks to a large range of solid systems with different dimensions. In fact, our graphene system is also a semi-metal, so calculations on graphene should already demonstrate the applicability of our methods for metal. In addition, the HEG system is often considered a model system for metal at corresponding electron densities. These calculations also supported the applicability of our methods to metallic solids.

However, as the reviewers suggested, we carried additional simulations of a normal three-dimensional solid metal, namely body-centered cubic lithium. We use a $2 \times 2 \times 2$ conventional body-centered cubic supercell with 16 atoms and 48 electrons. Below are our results.

Geometry of conventional bcc cell			
Atom	Position (\AA)	Lattice vector	Position (\AA)
Li 1	(0, 0, 0)	a_1	(3.436, 0, 0)
Li 2	(1.718,1.718,1.718)	a_2	(0, 3.436, 0)
		a_3	(0, 0, 3.436)

Result	Our network at Γ	Traditional QMC	Experiment
Total energy of unit cell	-15.04180(7) Ha	NA	NA
Cohesive energy	-1.168 eV/atom	-1.09(a), -1.57(b) eV/atom	-1.65 eV/atom

a. Sugiyama G, Zerah G and Alder B J 1989 Ground-state properties of metallic lithium, *Physica A* 156 144–168.

b. Yao G, Xu J G and Wang X W 1996 Pseudopotential variational quantum Monte Carlo approach to bcc lithium *Phys. Rev. B* 54 8393–8397.

We think our Γ point result is comparable to traditional QMC or experimental data. However, we also admit that there is a large finite-size error without a reliable correction scheme. We tried TABC techniques to reduce the finite-size error but encountered some problems for metal. **Overall**, it's well known that metal is the **most difficult** system for high accuracy approaches (such as QMC and CCSD), which requires much larger simulation size and more complicated techniques. Therefore, we think it's reasonable to leave it to our future work. We have added a note in the HEG section of revised text for this comment.

Assessment of comparable works/established literature

Comparing this paper to what is currently the state of the art in the field, there are a number of high impact publications. The best known paper is by Troyer in *Science* (Ref 9) and the original FermiNet paper was published in *Phys Rev Research* (Ref 10). The best case scenario of the technical development appears to be a more efficient version of FermiNet for solids and I would put this on a par with the original FermiNet development in terms of its specialism. In my view, this paper is better suited to a more specialized journal which would also allow the claims of the novelty of the network and any speed up to be better considered and reviewed.

Summary:

I think this is an interesting paper and it is important that it is published. However, my recommendation would be to seek publication in a journal that is well aligned with methods development.

If the applications are able to be significantly improved, this may find home in a more specialized Nature publishing group journal related to computation or materials science. I would be supportive of this paper being transferred to NPJ Computational Materials. While I do not think it would be suitable for publication in its current work at NPJ Computational Materials, I would be willing/able to provide further comments that -- if acted upon -- would

make it suitable. This would require the authors to show side by side comparisons with FermiNet (Ref 31) that show clear improvement. I also think the authors could collect more timing data and show improvement in that way, then they might reasonably try to submit to Nature Computational Science.

On the other hand, I think it would be immediately publishable in a journal such as the Journal of Chemical Theory and Computation, Phys Rev B, or Phys Rev Research.

We thank reviewer #3 for their efforts in evaluating our work. We would like to emphasize again that our work is **an independent work that has clear originality of methodological development, shows promising results for real solids and will inspire more works in the community.**

As has been clarified, this work is **not** an straightforward extension of Ref. 31. Our methods are developed independently, initially aimed at real solids. The application of our methods to HEG is done after our studies on solids. Although our HEG part overlaps with Ref. 31 (also Ref. 30), they are done in parallel before the release of these two relevant papers. We think it is reasonable to say our work is an extension of the original molecule FermiNet (Ref. 10). However, this extension is very significant, generalizing existing molecular neural networks to simulate solid systems. And the effectiveness of our extension is firmly supported in our manuscript. This work will attract a broad range of readers, including condensed matter physicists, chemists, materials scientists, and computational scientists. Therefore, we think Nature Communications is the most suitable journal for this work.

REVIEWER COMMENTS

Reviewer #2 (Remarks to the Author):

In this revised version of their manuscript, the authors have appropriately addressed the concerns of all the referees. I would like to thank the authors for their detailed answers to my comments. Now, I recommend the revised manuscript for publication in Nature Communications.

This is one additional comment: it would be helpful to add a reference (a textbook or a review article) for the generalised Bloch functions for many-body wavefunction, because the readers of Nature Communications are wide.

Reviewer #3 (Remarks to the Author):

I have reviewed the revised manuscript and the rebuttal from the authors. My objections and suggestions for improvement are as follows.

1. I do not disagree with much of what the authors have said in their rebuttal to my points about originality, but I believe they have missed the point.

My objection to the publication of this manuscript as it is currently being described are about the lack of an appropriate comparison to the work by Cassella et al. (arXiv:2202.05183). Casella et al. has priority within the community norms by appearing on arXiv well in advance of the manuscript that is under review here. While the work conducted by the authors was based on the molecular code (Pfau et al. Phys. Rev. Research 2, 033429 2020), it is always important for studies to compare their work to the state-of-the-art in the field as a whole. Omission of this comparison is disrespectful to the priority claim of Cassella.

I think that the advance that this study makes, when compared fairly to Cassella et al., is: (1) A different choice in how periodic boundary conditions are set up. (2) A lower degree of accuracy given the new data collected by the authors and presented in their rebuttal. These should be clearly stated or the authors should state what else they think the differences are.

After the first round of review, it would appear that the comparisons that were based on Pfau et al. were partially updated to Cassella et al. throughout. This should be completed. They should show what choices they made about periodic boundary conditions led to their program being able to calculate real solids, and what it is about Cassella et al. that makes it impossible for that algorithm to treat real solids.

2. I am pleased to see that the revised manuscript has toned down claims for the success of their method.

I also appreciate that the authors ran the requested calculations on $N=14$ and found their work was not accurate enough to reach FCIQMC accuracy. They have presented a reason for why their results are not as good as those of Cassella et al. All that remains is for this result to be merged into the main manuscript. Omitting results that are not complementary by putting them in the supplementary information makes the authors come across as though they are attempting to hide this outcome.

At the same time, the authors should also make comparisons to the FCIQMC data at $N=54$ for $r_s=0.5$ and 1.0 , by using these papers from Alavi and coworkers (Phys. Rev. B 85, 081103(R), 2012 and Phys. Rev. Research 3, 033072 2021).

3. The emphasis throughout the manuscript on real systems in an ab initio way is saying that you're able to treat the thermodynamic limit. When the manuscript states "These systems cover a wide range of interests, including materials dimension from one to three, electronic structure from metallic to insulating, and bonding type from covalent to ionic". This implies that long-range effects are being treated appropriately.

The benchmark results given in review on lithium provide the minimum coverage for all of these situations for realistic Hamiltonians, and these data should be included in the manuscript. It is a good idea for conclusions to include limitations of the current work, and this is a good place for the authors to discuss the challenges they have been having with metals and twist averaging.

I also think that a set of small benchmark systems should be provided so that multiple examples are available for each type of 3D solids, this would enable other authors to make better comparison with this method and bolster their claim of generality. It is also important that the authors include the HF values that these calculations are based on so that other authors can start from the same HF solution.

4. There is not enough provided on the method developed that the community would be able to independently reproduce the findings of the manuscript. The software is not well commented and has had comments removed from original authors from whom the code was taken.

Response to reviewers

Reviewer #2:

In this revised version of their manuscript, the authors have appropriately addressed the concerns of all the referees. I would like to thank the authors for their detailed answers to my comments. Now, I recommend the revised manuscript for publication in Nature Communications.

This is one additional comment: it would be helpful to add a reference (a textbook or a review article) for the generalised Bloch functions for many-body wavefunction, because the readers of Nature Communications are wide.

We thank reviewer 2 for his/her affirmation of our revision. The related reference of generalized Bloch functions are also added.

Reviewer #3:

I have reviewed the revised manuscript and the rebuttal from the authors. My objections and suggestions for improvement are as follows.

1. I do not disagree with much of what the authors have said in their rebuttal to my points about originality, but I believe they have missed the point.

My objection to the publication of this manuscript as it is currently being described are about the lack of an appropriate comparison to the work by Cassella et al. (arXiv:2202.05183).

Casella et al. has priority within the community norms by appearing on arXiv well in advance of the manuscript that is under review here. While the work conducted by the authors was based on the molecular code (Pfau et al. Phys. Rev. Research 2, 033429 2020), it is always important for studies to compare their work to the state-of-the-art in the field as a whole. Omission of this comparison is disrespectful to the priority claim of Cassella.

I think that the advance that this study makes, when compared fairly to Cassella et al., is: (1) A different choice in how periodic boundary conditions are set up. (2) A lower degree of accuracy given the new data collected by the authors and presented in their rebuttal. These should be clearly stated or the authors should state what else they think the differences are.

After the first round of review, it would appear that the comparisons that were based on Pfau et al. were partially updated to Cassella et al. throughout. This should be completed. They should show what choices they made about periodic boundary conditions led to their program being able to calculate real solids, and what it is about Cassella et al. that makes it impossible for that algorithm to treat real solids.

We thank the reviewer for the comments. In the revision, we made more comparisons with the works of Cassella et al. (Ref.[34] in our revision) and Wilson et al. (Ref.[33] in our revision), and added more discussions.

The first paragraph of section 2.2.4 has been rewritten.

Our revision in section 2.2.4:

"In addition to the solids containing nuclei, our computational framework can also apply straightforwardly to model systems such as homogeneous electron gas (HEG). HEG has been studied for a long time to understand the fundamental behavior of metals and electronic phase transitions [32]. Several seminal ab initio works have reported the energy of HEG at

different densities [25,26,32-35]. Recently two other works have extended neural network ansatz to study HEG [33,34]. Although our computational framework is independently designed for solids, the network structure between this work and Refs. [33,34] employ similar ideas. A different choice of periodic feature is made in our work, which could lead to a difference in the performance. Here, we make comparisons between these networks and ours on HEG, and observe consistent performance, which further proves the effectiveness of neural network based QMC works. In this section, we present the results calculated on a simple cubic cell containing 54 electrons in closed-shell configuration, the largest HEG system studied in this work (Fig. 2g). More results and comparisons with other works on smaller systems are discussed in subsection 3.1 and Supplementary Table 13."

We have added a new subsection in 3.1 with a new Fig. 5 discussing comparisons on the 14-electrons HEG system.

Our revision in section 3.1:

"In Refs. [33,34], neural networks are also used to simulate homogeneous electron gas system, employing a different choice of periodic feature function. In Fig. 5 we plot the correlation error computed on the 14-electrons HEG system, which can be compared with the results of other works. We can see that all three networks can go beyond BF-DMC level for high-density systems. For all systems tested, our correlation errors are about 2% with the TC-FCIQMC result as the reference [35], whereas the results of refs.[33,34] are within 1%. It is understandable that the networks of refs. [33,34] are specially designed for HEG systems, so slightly better accuracy can be achieved than our network. In their works, multiple phase factors $\exp(ik \cdot r)$ are used in the constructed orbitals, which improve the expressiveness of the network. In comparison, our network contains an additional exponential decay term, which simulates the attraction between atoms and electrons in solids containing nuclei (see Methods section for more details). Furthermore, the choice of periodic distance, as well as the domains of the constructed wavefunction (complex or real-valued), are also different in these three works, which may add differences to their performance. In the future, it would be interesting to combine the insights learned from these three works and design a better network ansatz for periodic systems."

Fig. 5 | Correlation error of 14-electrons HEG system at different r_s . Correlation error is defined as $[1 - (E - E_{\text{HF}})/(E_{\text{ref}} - E_{\text{HF}})] \times 100\%$. WAP-Net refers to Ref.[21] and FermiNet-HEG refers to Ref.[22]. BF-DMC results [27, 28] are displayed for comparison, and TC-FCIQMC data is used as reference [37].

2. I am pleased to see that the revised manuscript has toned down claims for the success of their method.

I also appreciate that the authors ran the requested calculations on $N=14$ and found their work was not accurate enough to reach FCIQMC accuracy. They have presented a reason for why their results are not as good as those of Cassella et al. All that remains is for this result to be merged into the main manuscript. Omitting results that are not complementary by putting them in the supplementary information makes the authors come across as though they are attempting to hide this outcome.

At the same time, the authors should also make comparisons to the FCIQMC data at $N=54$ for $r_s=0.5$ and 1.0 , by using these papers from Alavi and coworkers (Phys. Rev. B 85, 081103(R), 2012 and Phys. Rev. Research 3, 033072 2021).

We thank the reviewer for mentioning this point.

Additional comparisons to Refs.[21,22] are now included in our revision. See "network comparison" section and the response to comment 1.

TC-FCIQMC data at $N=54$ for $r_s=0.5$ and 1.0 are also included in Fig.2h of our revision(see below).

3. The emphasis throughout the manuscript on real systems in an ab initio way is saying that you're able to treat the thermodynamic limit. When the manuscript states "These systems cover a wide range of interests, including materials dimension from one to three, electronic structure from metallic to insulating, and bonding type from covalent to ionic". This implies that long-range effects are being treated appropriately.

Treating the thermodynamic limit (TDL) is indeed an important point when simulating real solids. In order to achieve TDL, we push our simulation size to the largest one ever reported in the neural network community (108 electrons). Standard finite-size error correction techniques are also adopted to reduce the long-range effects (as detailed throughout in different sections). The comparison between our work, state-of-the-art methods and experiment data are also positive. Besides, "Real solids" is also used in literature where similar scenarios exist, for example:

- George H. Booth, Andreas Grüneis, Georg Kresse, and Ali Alavi. Towards an exact description of electronic wavefunctions in **real solids**. Nature, 493:365–370, 2013.
- Nobuyuki Yoshioka, Wataru Mizukami, and Franco Nori. Solving quasi-particle band spectra of **real solids** using neural-network quantum states. Communications Physics, 4(1):1–8, 2021.

Considering these, we think it's **reasonable** to say that we study real solids in an *ab initio* way. In some text, the use of "real solids" is also necessary to differentiate from the general periodic system like HEG, which is also discussed in the manuscript. However, we understand the referee's concern and made revisions in a few places, where we think such a concern might occur.

Section 1:

"All the results demonstrate that our method can achieve accurate electronic structure calculations of real solids/periodic systems."

Section 2.2.4:

"In addition to the real solids containing nuclei, our computational framework can also apply straightforwardly to model systems such as homogeneous electron gas (HEG)."

The benchmark results given in review on lithium provide the minimum coverage for all of these situations for realistic Hamiltonians, and these data should be included in the manuscript. It is a good idea for conclusions to include limitations of the current work, and this is a good place for the authors to discuss the challenges they have been having with metals and twist averaging.

We thank the reviewer for this suggestion. Real metal systems are the most challenging ones for wavefunction theories, especially that they require large supercells and sophisticated finite size corrections. Such systems are indeed beyond the capability of current neural network methods. In light of the reviewer's comment, an additional "metallic lithium" section is now added in our revision as section 3.2. In the added section, we discuss the preliminary results obtained using our network on lithium and the limitations.

Our revision in section 3.2:

"We have also carried out preliminary calculations on metal lithium. Real metal system remains a notoriously difficult task for accurate wavefunction approaches [7,41-44]. The zero gap of metal leads to a discontinuity in the Brillouin zone integration. As a consequence, a significantly larger simulation cell is often required for metals than insulators to reach the thermodynamic limit. Shortcut approaches to simulate metals are proposed via employing a special twist angle [7, 43], which help to reduce the simulation size and finite-size error. We carried out calculations of lithium with a body-centered cubic (bcc) structure. A $2 \times 2 \times 2$ conventional cell of bcc-Li at Γ point is employed (see Supplementary Table 11). In Supplementary Table 12, we list the total energy and the cohesive energy computed. As expected, the error in cohesive energy of lithium with such a limited supercell is larger than non-metallic solids such as LiH, and further developments are desired to treat the large finite-size errors in metal."

I also think that a set of small benchmark systems should be provided so that multiple examples are available for each type of 3D solids, this would enable other authors to make better comparison with this method and bolster their claim of generality. It is also important that the authors include the HF values that these calculations are based on so that other authors can start from the same HF solution.

We also did calculations on a set of small benchmark systems as suggested by the referee. The results are listed in Supplementary Note.8. Corresponding HF values of each system are also listed in our supplementary file.

4. There is not enough provided on the method developed that the community would be able to independently reproduce the findings of the manuscript. The software is not well commented and has had comments removed from original authors from whom the code was taken.

We thank the reviewer for raising this issue. Detailed instructions of our neural network are now included in the "network structure" section to make it more self-contained.

As for the software, we have followed the standard open-source routine and properly reserved the copyright of original authors in each file. The removed comments are now added back. Some more detailed comments are also added now.

REVIEWERS' COMMENTS

Reviewer #3 (Remarks to the Author):

I'm happy that the authors have made all of the changes I have requested. Through this review process, the authors have been able to improve the manuscript so that it is comparing with the latest calculations within the community, ensuring that the work will have impact.

I have the following remaining suggestions:

1. I still think that the periodic versions that DeepMind have come up with need to be referenced at earlier/appropriate points in the manuscript. For example: "Here we propose a powerful periodic neural network ansatz for solids, which combines periodic distance features [20] with existed molecule neural networks [10]." and "However, how to apply such neural network ansatz for real solids in continuous space, i.e. how to apply periodic boundary conditions (PBC) in the neural network, and whether it can describe the long-range electron correlations in extended systems remain as open questions."

2. I'm pleased that the authors have made more calculations of bulk systems and the reasoning of these new sections generally looks sound. There are quite a few missing calculation details for the newly added solid state calculations. The authors would need to provide lattice shape, lattice parameters, supercell positions or k-points, basis sets, pseudopotentials, and offset k_s vector. These could be tabulated in the SI. But the easiest route is likely just the sharing of the version number of PySCF and outputs in a public repository like figshare (amongst others). This is the most reliable way to provide starting-points for others who may want to use your calculations and will improve the impact of your paper.

3. The inclusions of the authors around the algorithmic details of the manuscript are much improved. In particular, I want to highlight this sentence has been added to the paper which has the key intellectual merit of the work: "our network contains an additional exponential decay term, which simulates the attraction between atoms and electrons in solids containing nuclei (see Methods section for more details)." This is important because it has a physically motivated approximation that the authors are using to improve an algorithm. Focusing on this statement earlier in the manuscript would be beneficial.

For example, at this point: "the network structure between this work and Refs. [33, 34] employ similar ideas. A different choice of periodic feature is made in our work, which could lead to a difference in the performance", the algorithm should be summarized in terms of its physical characteristics. I don't think the difference in performance is all that important.

It's fine, even beneficial, to see different programs doing the same thing in the field. For instance: QChem and Gaussian. Two independent implementations doing their own thing is just fine.

4. For the 14 electron comparison, it would be useful for the authors to include in their SI table the comparison calculations used by Casella et al. which are from FCIQMC rather than TC-FCIQMC so that interested authors can make a like for like comparison. I think it's fine to leave the main manuscript figure as-is.

5. I am pleased that the authors have put the comments back in. I think the authors should push any remaining comments they have to the public version of the repository. I think that the most helpful comments are those that include descriptions of the input/output variables.

Response to reviewer

I'm happy that the authors have made all of the changes I have requested. Through this review process, the authors have been able to improve the manuscript so that it is comparing with the latest calculations within the community, ensuring that the work will have impact.

We would like to thank reviewer for his/her approval of our revision and many useful comments that have lead to the improvements of our manuscript.

1. I still think that the periodic versions that DeepMind have come up with need to be referenced at earlier/appropriate points in the manuscript. For example: "Here we propose a powerful periodic neural network ansatz for solids, which combines periodic distance features [20] with existed molecule neural networks [10]." and "However, how to apply such neural network ansatz for real solids in continuous space, i.e. how to apply periodic boundary conditions (PBC) in the neural network, and whether it can describe the long-range electron correlations in extended systems remain as open questions.".

We have added a sentence citing the two parallel works on HEG with periodic neural network in the Introduction section.

Our revision:

~~how to apply periodic boundary conditions (PBC) in the neural network,~~ (line 33)

In parallel to our work, Wilson et al. and Casella et al. [21,22] also developed periodic versions of neural networks to study the homogeneous electron gas system and obtained high-accuracy results. A more detailed comparison is discussed in the following sections. (line 54)

2. I'm pleased that the authors have made more calculations of bulk systems and the reasoning of these new sections generally looks sound. There are quite a few missing calculation details for the newly added solid state calculations. The authors would need to provide lattice shape, lattice parameters, supercell positions or k-points, basis sets, pseudopotentials, and offset k_s vector. These could be tabulated in the SI. But the easiest route is likely just the sharing of the version number of PySCF and outputs in a public repository like figshare (amongst others). This is the most reliable way to provide starting-points for others who may want to use your calculations and will improve the impact of your paper.

All the calculation details mentioned have already been included in Supplementary Note 8 and accoiated Supplementary Tables 15, 16. See also below. Moreover, the latest PySCF version

should work fine.

Supplementary Table 15 | Geometry of benchmark systems.

System	Structure	Atom	Position(Å)	lattice vector	Position(Å)
LiH	rock-salt	Li1	(0.0, 0.0, 0.0)	\mathbf{a}_1	(0.0, 2.0305, 2.0305)
		H	(2.0305, 2.0305, 2.0305)	\mathbf{a}_2	(2.0305, 0.0, 2.0305)
				\mathbf{a}_3	(2.0305, 2.0305, 0.0)
Li	bcc	Li1	(0.0, 0.0, 0.0)	\mathbf{a}_1	(3.436, 0.0, 0.0)
		Li2	(1.718, 1.718, 1.718)	\mathbf{a}_2	(0.0, 3.436, 0.0)
				\mathbf{a}_3	(0.0, 0.0, 3.436)
Be	hexagonal	Be1	(1.1299, 0.6524, 2.6774)	\mathbf{a}_1	(2.2598, 0.0, 0.0)
		Be2	(0.0, 1.3047, 0.8925)	\mathbf{a}_2	(-1.1299, 1.9571, 0.0)
				\mathbf{a}_3	(0.0, 0.0, 3.5699)
C	diamond	C1	(0.0, 0.0, 0.0)	\mathbf{a}_1	(0.0, 1.7869, 1.7869)
		C2	(0.8934, 0.8934, 0.8934)	\mathbf{a}_2	(1.7869, 0.0, 1.7869)
				\mathbf{a}_3	(1.7869, 1.7869, 0.0)
C	graphene	C1	(1.421, 0.0, 0.0)	\mathbf{a}_1	(2.1315, -1.2306, 0.0)
		C2	(2.842, 0.0, 0.0)	\mathbf{a}_2	(2.1315, 1.2306, 0.0)
				\mathbf{a}_3	(0, 0, 52.9177)

Supplementary Table 16 | Calculated energy of selected small systems. A $1 \times 1 \times 1$ cell at Γ point is used. HF results are calculated in cc-pVDZ basis. Energies are given in Hartree. N_e denotes number of electrons.

System	Structure	Net	HF	N_e
LiH	rock-salt	-8.5165(2)	-8.4512	4
Li	bcc	-15.34486(1)	-14.38643	6
Be	hexagonal	-30.2416(1)	-30.0752	8
C	diamond	-75.4009(2)	-74.9784	12
C	graphene	-76.0350(2)	-75.5779	12

3. The inclusions of the authors around the algorithmic details of the manuscript are much improved. In particular, I want to highlight this sentence has been added to the paper which has the key intellectual merit of the work: "our network contains an additional exponential decay term, which simulates the attraction between atoms and electrons in solids containing nuclei (see Methods section for more details)." This is important because it has a physically motivated approximation that the authors are using to improve an algorithm. Focusing on this statement earlier in the manuscript would be beneficial.

For example, at this point: "the network structure between this work and Refs. [33, 34] employ similar ideas. A different choice of periodic feature is made in our work, which could lead to a difference in the performance", the algorithm should be summarized in terms of its physical characteristics. I don't think the difference in performance is all that important.

It's fine, even beneficial, to see different programs doing the same thing in the field. For instance: QChem and Gaussian. Two independent implementations doing their own thing is just fine.

We thank the reviewer for this suggestion. Statement mentioned above has been reorganized to manifest the underlying physical characteristics of these algorithms:

Our revision:

~~A different choice of periodic feature is made in our work, which could lead to a difference in the performance.~~ Different physics-inspired envelope functions and periodic features are used in these works, which suit the features of solids and homogeneous electron gas respectively.

(line 182)

4. For the 14 electron comparison, it would be useful for the authors to include in their SI table the comparison calculations used by Casella et al. which are from FCIQMC rather than TC-FCIQMC so that interested authors can make a like for like comparison. I think it's fine to leave the main manuscript figure as-is.

Mentioned i-FCIQMC data is added now to our supplementary Table 13.

5. I am pleased that the authors have put the comments back in. I think the authors should push any remaining comments they have to the public version of the repository. I think that the most helpful comments are those that include descriptions of the input/output variables.

We thank reviewer for mentioning this point. We have double checked our comments, and we think the descriptions of input/output variables are complete.